# *Perilla frutescens* Leaf Extract Attenuates Vascular Dementia-Associated Memory Deficits, Neuronal Damages, and Microglial Activation

**Hyun-Bae Kang [1],†, Shin-Hye Kim [2],†, Sun-Ho Uhm [1], Do-Kyung Kim [1], Nam-Seob Lee [1], Young-Gil Jeong [1], Nak-Yun Sung [3], Dong-Sub Kim [3], In-Jun Han [3], Young-Choon Yoo [4] and Seung-Yun Han [1],***

[1] Department of Anatomy, College of Medicine, Konyang University, Daejeon 35365, Korea; 971801@daum.net (H.-B.K.); Always703@naver.com (S.-H.U.); dokyung@konyang.ac.kr (D.-K.K.); nslee@konyang.ac.kr (N.-S.L.); ygjeong@konyang.ac.kr (Y.-G.J.)

[2] Department of Biological Sciences, Korea Advanced Institute of Science and Technology (KAIST), Daejeon 34141, Korea; shinhye@kaist.ac.kr

[3] Division of Natural Product Research, Korea Prime Pharmacy Co., Ltd., Jeonnam 58144, Korea; ny.sung@koreaprime.co.kr (N.-Y.S.); ds.kim@koreaprime.co.kr (D.-S.K.); ij.han@koreaprime.co.kr (I.-J.H.)

[4] Department of Microbiology, College of Medicine, Konyang University, Daejeon 35365, Korea; yc_yoo@konyang.ac.kr

\* Correspondence: jjzzy@konyang.ac.kr; Tel.: +82-042-600-8616

† These authors contributed equally to this work.

**Abstract:** Vascular dementia (VaD) is characterized by a time-dependent memory deficit and essentially combined with evidence of neuroinflammation. Thus, polyphenol-rich natural plants, which possess anti-inflammatory properties, have received much scientific attention. This study investigated whether *Perilla frutescens* leaf extract (PFL) exerts therapeutic efficacy against VaD. Sprague Dawley rats were divided into five groups: SO, sham-operated and vehicle treatment; OP, operated and vehicle treatment; PFL-L, operated and low-dose (30 mg/kg) PFL treatment; PFL-M, operated and medium-dose (60 mg/kg) PFL treatment; and PFL-H, operated and high-dose (90 mg/kg) PFL treatment. Two-vessel occlusion and hypovolemia (2VO/H) were employed as a surgical model of VaD, and PFL was given orally perioperatively for 23 days. The rats underwent the Y-maze, Barnes maze, and passive avoidance tests and their brains were subjected to histologic studies. The OP group showed VaD-associated memory deficits, hippocampal neuronal death, and microglial activation; however, the PFL-treated groups showed significant attenuations in all of the above parameters. Using lipopolysaccharide (LPS)-stimulated BV-2 cells, a murine microglial cell line, we measured PFL-mediated changes on the production of nitric oxide (NO), TNF-$\alpha$, and IL-6, and the activities of their upstream MAP kinases (MAPKs)/NF$\kappa$B/inducible NO synthase (iNOS). The LPS-induced upregulations of NO, TNF-$\alpha$, and IL-6 production and MAPKs/NF$\kappa$B/iNOS activities were globally and significantly reversed by 12-h pretreatment of PFL. This suggests that PFL can counteract VaD-associated structural and functional deterioration through the attenuation of neuroinflammation.

**Keywords:** *Perilla frutescens*; vascular dementia; hippocampal neuron; neuroinflammation

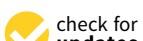



## 1. Introduction

Vascular dementia (VaD) is a progressive neurodegenerative disorder that affects cognitive abilities and is caused by cerebrovascular accidents (CVAs) [1]. VaD accounts for approximately 17–20% of all dementia cases, rendering it the second leading subtype of dementia after Alzheimer's disease (AD), and is prevalent among elderly people [2]. The most important pathology induced by CVA is cerebral infarction, a discrete region of tissue that encounters significant demise in cerebral blood flow (CBF) [3]. Due to the deprivation of oxygen and nutrients, the infarct tissue becomes susceptible to reactive oxygen species (ROS)-associated oxidative damage and subsequent neuroinflammation [4].

Experimental evidence focusing on the role of neuroinflammation in VaD suggests that microglia obtain an M2 phenotype, which gradually changes into a proinflammatory M1 phenotype, also known as "activated microglia", in the area surrounding infarcts [5,6]. The pathologic mechanism, which is associated with ROS overproduction, can stimulate the release of proinflammatory mediators from activated microglia, and this impact can be exacerbated by an increase in blood–brain barrier permeability, which empowers the extravasation of proinflammatory mediators, for example, nitric oxide (NO), interleukin (IL)-6, and tissue necrosis factor (TNF)-$\alpha$ [7,8]. In addition, the accompanying activation of inducible NO synthase (iNOS) and mitogen-activated protein kinases (MAPKs), such as p38, extracellular signal-regulated kinase (ERK), and c-Jun N-terminal kinases (JNK), as well as nuclear translocation of nuclear factor kappa B (NFκB) in peri-infarct microglia, all converge to eventual tissue death in various cerebral regions, including the hippocampus, the center of sensorimotor or cognitive capacities [9,10].

Considering the key roles of oxidative stress and neuroinflammation in VaD pathology described above, screening of antioxidant or anti-inflammatory agents that could be exogenously supplemented is of great scientific interest. Exogenous antioxidants, especially polyphenols, are commonly derived from medicinal plants, which exhibit a wide range of biological effects including anti-inflammatory [11], antibacterial [12], antiviral [13], anti-aging [14], and anticancer properties [15]. Among the polyphenol-rich medicinal plants, *Perilla frutescens* (L.) Britt. is an annual herbal organism that belongs to the mint family Lamiaceae [16]. It is widely cultivated in Asian countries, where it has been used as a valuable source of traditional medicinal uses [17]. In total, 271 phytochemical compounds, including polyphenols, such as scutellarin, apigenin-7-O-glucuronide, and rosmarinic acid, have been identified in their seeds, stems, and leaves [18].

The efficacy of *P. frutescens* polyphenols, as either isolated compounds or in the form of crude leaves, seeds, and stem extracts, has been tested experimentally for various central nervous system disorders, such as trauma [19], ischemic injury [20], and AD [21,22]. However, despite accumulating evidence regarding the neuroprotective efficacies of *P. frutescens*, to the best of our knowledge, the potential therapeutic effect on VaD pathology has not yet been demonstrated. In this study, we investigated the therapeutic potential of *P. frutescens* leaf extract (PFL) for VaD, and the underlying mechanism focused on the microglial MAPKs/NFκB/iNOS dynamics, which is proposed as a surrogate of microglial activation using an experimental VaD model in vivo and a BV-2 cell line in vitro.

## 2. Materials and Methods

### 2.1. Chemicals and Antibodies

Lipopolysaccharide (LPS; *Escherichia coli* O55:B5), dimethyl sulfoxide (DMSO), scutellarin, apigenin-7-O-glucuronide, rosmarinic acid, trifluoroacetic acid (TFA), 3-(4,5-Dimethylthiazol-2-yl)-2,5-diphenyl tetrazolium bromide (MTT), bicinchoninic acid (BCA) kit, and cresyl-violet (C-V) powder were purchased from Sigma (St. Louis, MO, USA). Dulbecco's modified Eagle's medium (DMEM), penicillin-streptomycin (P/S) mixture, and fetal bovine serum (FBS) were obtained from Gibco (Grand Island, NJ, USA). Enzyme-linked immunosorbent assay (ELISA) kits for measuring NO, TNF-$\alpha$, and IL-6 were purchased from R&D Systems (Minneapolis, MN, USA). Primary and secondary antibodies were purchased from Cell Signaling (Danvers, MA, USA). Tertiary antibodies (avidin-biotin complex kit) and 3,3'-diaminobenzidine (DAB) were purchased from Vector Laboratories (Burlingame, CA, USA). Cell lysis buffer was purchased from iNtRON (PRO-PREP™; Seongnam, Korea). The cell compartment kit was purchased from QIAGEN (Qproteome; Hilden, Germany). An enhanced chemiluminescence kit was purchased from Merck (Immobilon™; Darmstadt, Germany).

### 2.2. Preparation of Perilla Frutescens Leaf Extract

Fresh *Perilla frutescens* leaves were harvested from the private farm of Korea Prime Pharmacy CO., LTD. (Jeonnam, Korea) at August in 2019. Hot air drying was performed

at 50 °C using a hot air-dry oven (PURIVEN 150; Cryste, Seoul, Korea) until there was no change in moisture (10%) anymore. Two kilograms of dried Perilla leaves were mixed with 36 L of 70% EtOH, and extracted with reflux condensation for 4 h at 75 °C. Undissolved residues were removed by filtration through a Whatman 42 filter paper (Clifton, NJ, USA), followed by filtration through a 0.45 μm membrane filter (Millipore, Billerica, MA, USA). The filtrate was evaporated under vacuum and freeze-dried. The dried extracts (PFL) were stored at 4 °C until analysis. The residue was lyophilized (Freeze dryer FD8508; IlShinBioBase CO. LTD., Gyeonggi, Korea) and the amount of PFL was 206 g (extraction yield: 10.3%). The lyophilized powder was stored at 4 °C until use.

### 2.3. High-Performance Liquid Chromatography (HPLC)

The HPLC system (LC-20AD; Shimadzu, Kyoto, Japan) using an SPD-M20A diode array detector and YMC Triart C18 column (4.6 × 250 mm, 5 μm) was used to confirm the presence of scutellarin, apigenin-7-O-glucuronide, and rosmarinic acid in PFL, as reported in a previous study [23]. The mobile phase consisted of solvents A and B. Solvent A was 0.1% TFA in water, and solvent B was 0.1% TFA in acetonitrile. The gradient was 0–3 min, 1–10% B; 3–20 min, 10–50% B; 20–20.1 min, 50–100% B; 20.1–24 min, 50–100% B; 24–24.1 min, 100–10% B; and 24.1–30 min, 10% B. Run time was 30 min, with a flow rate of 1.0 mL/min. The phenolic compounds were identified by the retention time and UV spectrum of the standard measured from the peak area at 254 nm. The concentration was calculated by comparing the peak areas of the samples with the calibration curve of the standards. Stock standard solutions of scutellarin, apigenin-7-O-glucuronide, and rosmarinic acid at a concentration of 1 mg/mL in 70% aqueous ethanol were prepared. Finally, they were diluted into seven standard concentrations of 1.95, 3.91, 7.81, 15.63, and 31.30 μg/mL, respectively, which were prepared for the construction of the calibration curves. Fifty milligrams of the PFL EtOH extract were accurately weighed and then extracted with 50 mL of 70% aqueous EtOH at room temperature through sonication, three times, with intervals of 10 min each. The supernatant was filtered and then determined by HPLC. HPLC chromatograms of PFL compared with those of mixtures of the three standard compounds, and the quantified amount of each constituent, are presented in Figure 1.

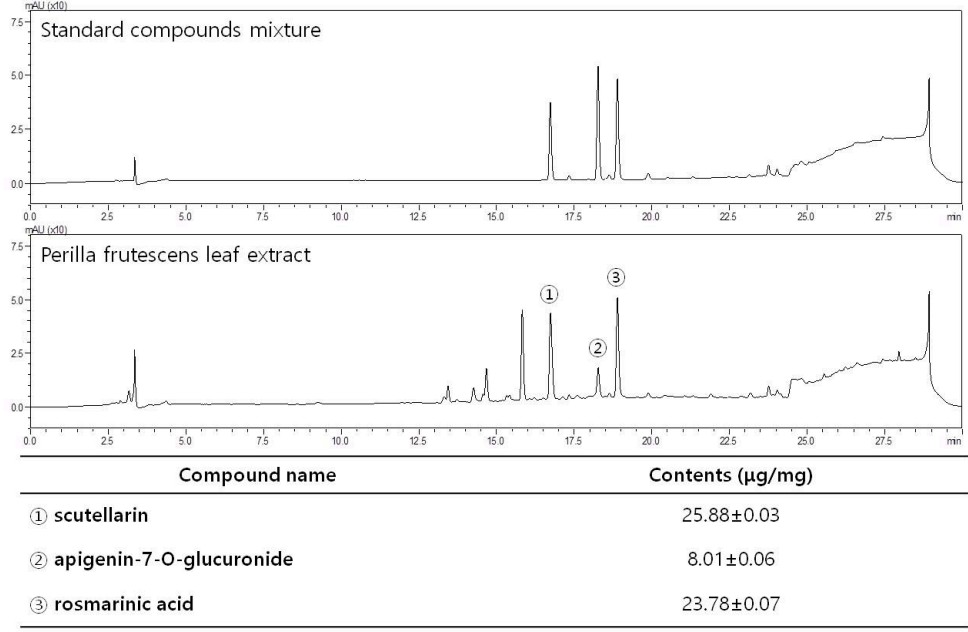

| Compound name | Contents (µg/mg) |
|---|---|
| ① scutellarin | 25.88±0.03 |
| ② apigenin-7-O-glucuronide | 8.01±0.06 |
| ③ rosmarinic acid | 23.78±0.07 |

**Figure 1.** HPLC chromatograms of a standard mixture of scutellarin, apigenin-7-O-glucuronide, and rosmarinic acid, and Perilla frutescens leaf extract obtained using a YMC Triart C18 column monitored at 254 nm of peak area, 30 min of run time, 1.0 mL/min of flow rate. Peaks of scutellarin, apigenin-7-O-glucuronide, and rosmarinic acid are indicated with circled numbers "1", "2", and "3", respectively.

### 2.4. Animals

A total of 50 male Sprague Dawley rats (8 weeks, 200–250 g) were obtained from Samtako (Osan, Korea) and kept in an environmentally controlled room for 7 days at a constant temperature (21–23 °C) and relative humidity (45–65%), under a 12 h light/dark cycle. Water and food were available *ad libitum*. Rats were housed in groups of 5 per cage, and were randomly assigned to different groups for all experiments. The experiments were executed in accordance with the Guide for the Care and Use of Laboratory Animals (National Institutes of Health publication, 8th Edition, 2011) [24]. Animal experiments in this study were approved by the Institutional Animal Care and Use Committee (IACUC) of Konyang University (Daejeon, Korea; approved protocol number: P-20–31-A-01).

### 2.5. Experimental Plan

All rats were randomly assigned to four groups, and treated differently, as follows: with distilled water used as vehicle ($n = 20$), with a low dose of PFL (30 mg/kg; $n = 10$; PFL-L), with a medium dose of PFL (60 mg/kg; $n = 10$; PFL-M), and with a high dose of PFL (90 mg/kg; $n = 10$; PFL-H). The selection of a high dose of PFL was based on the dose conversion protocols that can be found elsewhere [25]. Just prior to operation, the groups with vehicle were further divided into two groups ($n = 10$) as follows: with sham-operation (SO), and with operation (VEH), which eventually yielded five groups, titled: "SO", "VEH", "PFL-L", "PFL-M", and "PFL-H". The PFL was dissolved in a vehicle to obtain a final volume of 1 mL, administered intraorally once a day for 14 days before the surgery and an additional 9 days after surgery. From postoperative day (POD) 3 to POD 9, all groups underwent three different behavioral tests in sequential order (see Sections 2.7–2.9). After finishing all the tests, the rats were sacrificed for tissue sampling (see Section 2.10). In this experiment, group allocation, outcome assessment, and data analysis were performed by the authors blinded to the study. The animal experimental procedures were performed in a way that adhered to the Animal Research: Reporting of in vivo Experiments (ARRIVE) guidelines.The time flow of the experimental plan in this study is depicted in Figure 2a.

### 2.6. Surgery

Two-vessel occlusion and hypovolemia (2VO/H) were employed to establish a rat model of VaD using a slightly modified version of a previously described method by Sanderson and Wider [26]. For this, the rats were anesthetized with 3% isoflurane in 70% $N_2O$ and 30% $O_2$, and this state was maintained throughout the surgery at a level of 1.5~2% isoflurane. The rectal temperature was controlled at 37 °C with a heating pad during the entire surgery. The surgical procedure for 2VO/H is illustrated in Figure 2b. The left femoral artery was isolated and catheterized with a PE-50 catheter connected to a heparinized syringe to allow the future withdrawal of blood. The two common carotid arteries (CCAs) were exposed and permanently ligated with a 4–0 nylon suture. Following this, blood was withdrawn via a catheterized femoral artery at a constant speed (1 mL/min) to cause hypovolemia. When the relative CBF value, as measured by Laser-Doppler flowmetry (Periflux 5000; Perimed AB, Järfälla, Sweden), reached below 20% of the baseline, the blood withdrawal was stopped for 8 min to maintain the ischemic period (Figure 2c). During this period, a syringe filled with blood was kept in a water bath maintained at 37 °C. At the end of the ischemic period, the blood was reinfused at a constant speed (1 mL/min), and the relative CBF value returned to nearly 60% of the baseline value. After the closure of the surgical wound, the rats were maintained in their home cage.

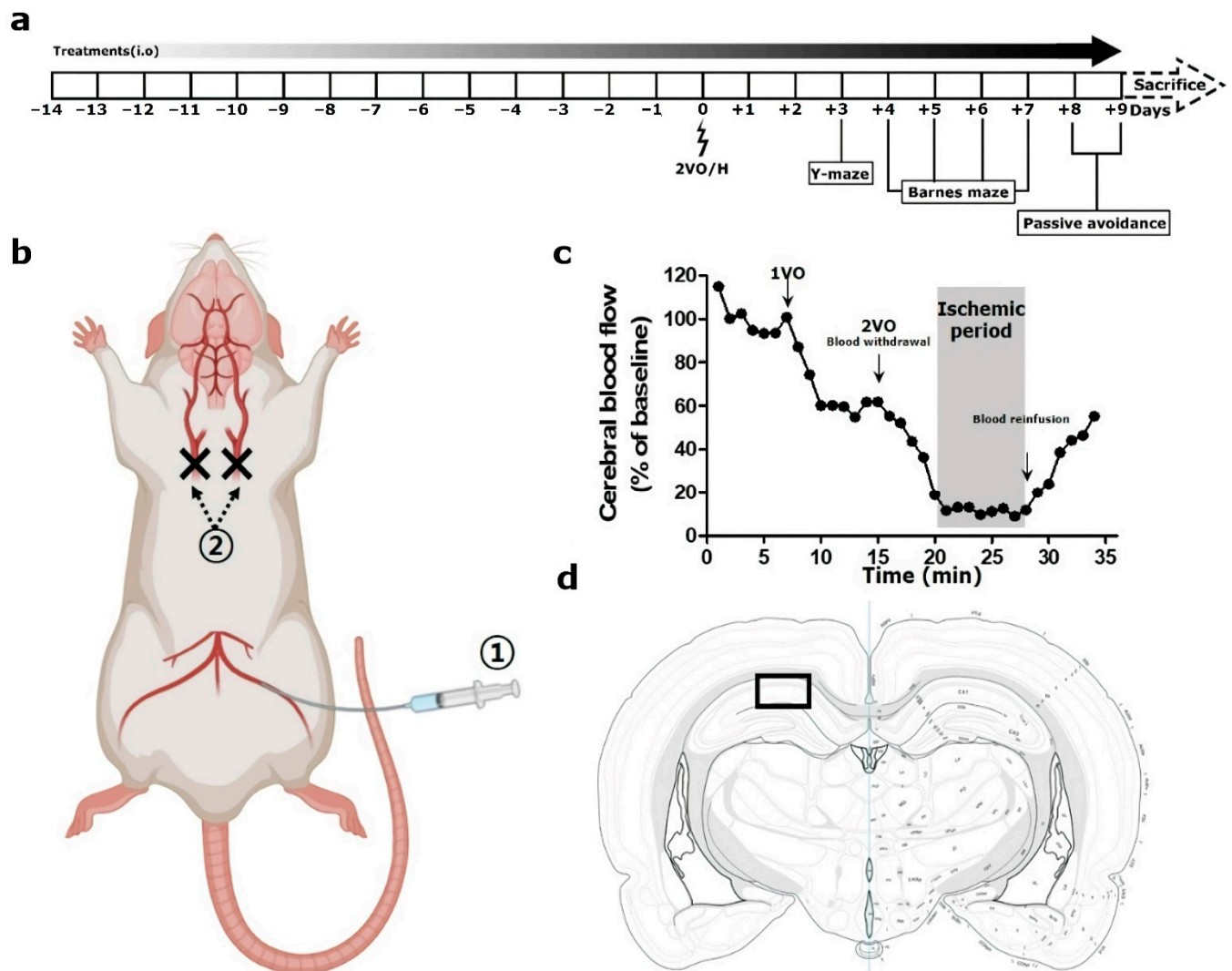

**Figure 2.** Graphical illustration of the in vivo experiment setup. (**a**) Schematic overview of the in vivo experiment, (**b**) surgical procedure, (**c**) a representative Doppler flowmetry traced for the entire operation time, and (**d**) the area of interest for histologic analyses in this study are presented. In (**b**), circled "1" and "2" represent the steps of femoral artery catheterization and ligatures of both common carotid arteries in the 2VO/H operation, respectively. In (**d**), a rectangular box indicates a 300 μm-width area in hippocampal Cornu ammonis 1, which was used for histologic analyses.

### 2.7. Y-Maze Test

On POD 3, rats were initially introduced to the center of the matte black plastic maze, with three arms, each with a length of 50 cm, width of 15 cm, and height of 30 cm, at 120° intervals. The sequence and number of arm entries were monitored for an 8-min period. The number of total arm entries also served as an indicator of locomotor activity. To calculate the percentage of spontaneous alternation behavior, the following formula was used: spontaneous alternation (%) = [(number of alternations)/(total arm entries − 2)] × 100. After each rat was tested, the maze was cleaned using 70% EtOH.

### 2.8. Barnes Maze Test

From POD 4 to 7, the Barnes maze test was performed. For this, a 100-cm-high and 122-cm-diameter circular maze was used. There are 20 holes located around the perimeter with a black escape box (20 × 15 × 12 cm) placed under one of these. The task was divided into two sessions: trial test session and probe test session. For the trial test session, which

was executed on POD 4, 5, and 6, the rats were placed on a table and given 120 s to find and enter the escape box, with bright lights as aversive stimuli. Each animal performed one trial per day. For the probe test session, which was executed on POD 7, the escape box was removed, and the time spent per 120 s in the quadrant where the escape box was originally located was recorded. The explorative behavior of the rats during the two sessions was recorded and analyzed with the aid of a video camera connected to an EthoVision XT9 system (Noldus, Wageningen, the Netherlands). After each rat was tested, the maze was cleaned using 70% EtOH.

### 2.9. Passive Avoidance Test

On POD 8 and 9, a passive avoidance test was performed. The test apparatus was equipped with two chambers: an illuminated chamber and a dark chamber, each measuring $25 \times 20 \times 25$ cm. A lamp (50 W) was used as an aversive stimulus in the illuminated chamber. Each test involved two separate sessions: a trial session and a test session. During the trial session, which was executed on POD 8, the rats were initially placed in the illuminated chamber. Once the rat had entered the dark compartment, the "trial latency" was measured using a stopwatch. This measurement was triplicated, and an electrical shock (0.5 mA) was delivered for 3 s through stainless steel rods at the last entry of the rat to the dark chamber. A test session was performed 24 h after the trial session, and the "escape latency" to re-enter the dark chamber from the illuminated chamber was measured up to 180 s. After each rat was tested, the chambers were cleaned using 70% EtOH.

### 2.10. Tissue Processing and Cresyl Violet Stain

After finishing all the behavioral tests, the rats were euthanized with an intraperitoneal injection of 100 mg/kg thiopental (Trapanal; Nycomed, Konstanz, Germany) and their brains was removed after transcardial perfusion with 100 mL of 4% paraformaldehyde diluted in phosphate-buffered saline (PBS). The isolated brains were post-fixed, dehydrated, paraffin-embedded, and serially sectioned at a 5-μm thickness using a microtome (RM2255; Leica, Nussloch, Germany). Two slides, randomly selected from the hippocampus-bearing tissue collections ($-3.0$ to $-4.0$ mm from the bregma in stereotaxic coordinates) of each rat, were deparaffinized, hydrated, and washed twice in PBS. The slides were stained with 0.1% cresyl violet (C-V) solution. The hippocampal Cornu ammonis 1 (CA1) regions were photographed at $200\times$ magnification using a digital camera connected to a light microscope (DM4; Leica). From the images, the number of intact neurons bearing clear nuclei and large cell bodies in the area of interest (CA1 subfield within a 300 μm width; depicted in Figure 2d as a rectangular box) were counted and averaged.

### 2.11. Immunohistochemistry

The two deparaffinized slides randomly chosen from each rat were incubated with rabbit anti-glial fibrillary acidic protein (GFAP) and rabbit anti-ionized calcium-binding adapter molecule 1 (Iba-1), each diluted in PBS to a ratio of 1:200, in a humid chamber for 24 h at 4 °C. After three washes in PBS, each slice was incubated with biotinylated anti-rabbit IgG antibodies, which were diluted in PBS to a ratio of 1:250, in a humid chamber for 2 h at 22–24 °C. After 3 washes in PBS, the slides were further incubated with an avidin-biotin complex, which was diluted in PBS to a ratio of 1:250, for 1 h at 22–24 °C. The resulting immunoreactivities turned brown in color after the addition of DAB, which was used as a chromogen. After mounting, the area of interest (CA1 subfield within a 300 μm width) was photographed at $200\times$ magnification using a digital camera connected to a light microscope. In each photograph, while the GFAP-immunoreactive ( $^+$ ) densities were quantified using Image J (v1.49, National Institutes of Health, Bethesda, MD, USA), the number of Iba-1$^+$ cells were counted manually. The results were averaged per group.

### 2.12. Cell Cultures and Cell Viability Assay

The murine BV-2 cells (ATCC, Rockville, MD, USA) were maintained in DMEM supplemented with 10% FBS and 1% P/S in a $CO_2$ incubator maintained at 37 °C. To determine the appropriate dosage of PFL treatments in this study, the cells were seeded in a 96-well plate at a density of $1 \times 10^4$ cells/well and incubated for 24 h at 37 °C. The cells were then treated with PFL (0, 50, 100, and 200 µg/mL) for 12 h and subsequently stimulated with or without LPS (1 µg/mL) for an additional 24 h. Cell viability was determined based on the MTT reduction. The MTT solution was created by diluting MTT in PBS at a final concentration of 0.5 mg/mL. The MTT solution was added to each well, and the plate was incubated for 4 h at 37 °C. After incubation, the culture medium was removed, and the resulting insoluble formazan crystals were dissolved in 100 µL of DMSO. The absorbance was measured at 540 nm in an ELx800UV microplate reader (BioTek Instruments, Winooski, VT, USA), and the data were expressed as the percentage of viable cells relative to untreated controls.

### 2.13. ELISA

Cells ($5 \times 10^4$ cells/well) were pretreated with PFL (0, 50, 100, and 200 µg/mL) for 12 h, followed by stimulation with 1 µg/mL LPS for 24 h. Then, 100 µL of cultured media was collected and stored at −75 °C. The culture media were mixed with different reagents contained in NO, TNF-$\alpha$, and IL-6 ELISA kits according to the manufacturer's instructions. The absorbance of the mixtures was measured with a microplate reader at 540 nm in a microplate reader, and each individual component was quantitatively determined by comparison to the standard curve.

### 2.14. Western Blot

Cells ($1 \times 10^7$ cells/well in 12-well plates) were pretreated with PFL (0, 50, 100, and 200 µg/mL) for 12 h, followed by stimulation with 1 µg/mL LPS for 24 h. The cell pellets were collected by centrifugation, resuspended in PRO-PREP$^{TM}$ lysis buffer, and centrifuged. Protein concentrations in the supernatant fractions were determined using a BCA assay kit. Then, 30 µg aliquots of protein were subjected to electrophoresis on 10% sodium dodecyl sulfate-polyacrylamide gels and transferred onto polyvinylidene fluoride membranes (Bio-Rad, Hercules, CA, USA). The membranes were incubated in a 5% skim milk solution and then with antibodies against phosphorylated (*p*-) p38, p38, *p*-extracellular signal-regulated kinase (*p*-ERK), ERK, *p*-c-Jun N-terminal kinase (*p*-JNK), JNK, and iNOS. For blotting of NF$\kappa$B, the nuclear fraction was obtained using a Qproteome cell compartment kit, as described by the manufacturer. *β*-actin and LaminB1 were used as internal controls of cellular homogenates and nuclear fractions, respectively. The membranes were washed with PBS containing 0.1% Tween-20, incubated with the horseradish peroxidase-conjugated secondary antibody for 1 h, and the resulting bands were identified with an enhanced chemiluminescence kit. The band intensity was measured using ImageJ and normalized to that of *β*-actin or LaminB1. Data were expressed as fold changes from the control values and collected from at least three independent experiments.

### 2.15. Statistics

All data are presented as the mean $\pm$ standard error of the mean (SEM). Sample sizes were determined by power calculations based on our previous studies and preliminary results (power, 90%; $\alpha = 0.05$). The normal distribution of the data was confirmed by the Shapiro–Wilk test. Comparisons of the data from the different groups were performed with the one-way analysis of variance (PASW Statistics version 18; SPSS Inc., Chicago, IL; RRID:SCR_002865) test. Values of $p < 0.05$ were considered statistically significant.

## 3. Results

### 3.1. PFL Attenuates Memory Impairment In Vivo in an AD Rat Model

With the aim of assessing the quality of aqueous PFL, the existence and quantification of polyphenolic constituents were analyzed using HPLC. In accordance with previous reports [23], three major bioactive constituents; scutellarin, apigenin-7-O-glucuronide, and rosmarinic acid, were detected in PFL (Figure 1). Given that three polyphenolic compounds are well known for their antioxidant- [14] and anti-inflammatory properties [11], we could establish a hypothesis regarding the potential protective effects on in vivo VaD models. First, in this study, we investigated whether PFL can attenuate memory impairment, which is an essential feature in VaD patients. As presented in the schematic overview (Figure 2a), we assessed the effects of 23 (preoperative 14 and postoperative 9) days of intake of PFL on the memory performance of the VaD rats. As shown in Figure 2b,c, in general, the VaD rats were reliably produced by 2VO/H operation with the guidance of Doppler flowmetry. However, one rat of the VEH group died immediately after the operation, and one rat of the PFL-L group failed to reach below 20% of the baseline in CBF during the ischemic period, and were therefore excluded. From POD 3, the rats were subjected to three different behavioral analyses, that is, Y-maze, Barnes, and passive avoidance tests in a sequential manner. The Y-maze test results showed that the VEH group exhibited significant impairments in memory, as indicated by the decrease in spontaneous alternation (*** $p < 0.001$ vs. SO; Figure 3a). However, the values were significantly spared in the PFL-M and PFL-H groups when compared with the VEH group ($52.2 \pm 4.4$ and $58.0 \pm 6.6$ vs. $37.5 \pm 5.3$; # $p < 0.05$ and ## $p < 0.01$ vs. VEH, respectively), while the PFL-L group showed no improvement. Neither the operation nor PFL treatment significantly affected locomotor activity, as measured by the number of total arm entries (Figure 3b). The Barnes maze results demonstrated that all the groups moved with identical distances to explore the escape platform on the first day of the trial test session (Figure 3d). However, on the second and third day of the session, the VEH group moved over a longer distance to find the platform when compared with the SO group (*** $p < 0.001$ vs. SO; Figure 3c,d). Although the differences in these values between the PFL-L/PFL-M groups and the VEH group were insignificant, the PFL-H group showed an apparent decrease in the moving distance to the goal on the second and third day of the session (# $p < 0.05$ vs. VEH). Neither the operation nor PFL treatment induced significant changes in motor performance, as determined by the moving velocity (data not shown). During the probe test session, the VEH group spent less time in the target quadrant where the escape platform was formerly located (*** $p < 0.001$ vs. SO; Figure 3e,f), suggesting that 2VO/H triggered memory deficits in rats. Interestingly, this deficit was significantly prevented in the PFL-H ($51.1 \pm 1.2$ vs. $26.9 \pm 7.3$; ## $p < 0.01$ vs. VEH). Although there were no statistically significant differences between the PFL-L and PFL-M groups over the VEH group, the time spent in the target quadrant was higher in the PFL-H group than in the PFL-L group (vs. $29.0 \pm 2.1$; §§ $p < 0.01$), indicating the existence of dose dependency. Finally, passive avoidance test results showed that the VEH group developed a significant impairment in retention memory compared to the SO group, as shown by the significantly lower escape latencies (*** $p < 0.001$; Figure 3h) in the test sessions. However, the latencies of the PFL-M and PFL-H groups were significantly longer than those of the VEH group ($109.1 \pm 7.4$ and $159.2 \pm 6.8$ vs. $50.3 \pm 6.1$; ## $p < 0.05$ and ### $p < 0.001$, respectively), and these changes were dose dependent (vs. $76.1 \pm 7.1$ in PFL-L; §§ $p < 0.01$). Neither the operation nor PFL treatments affected the trial latencies assessed during the training sessions (Figure 3g). Altogether, these results suggest that PFL intake can attenuate memory impairment in VaD rats.

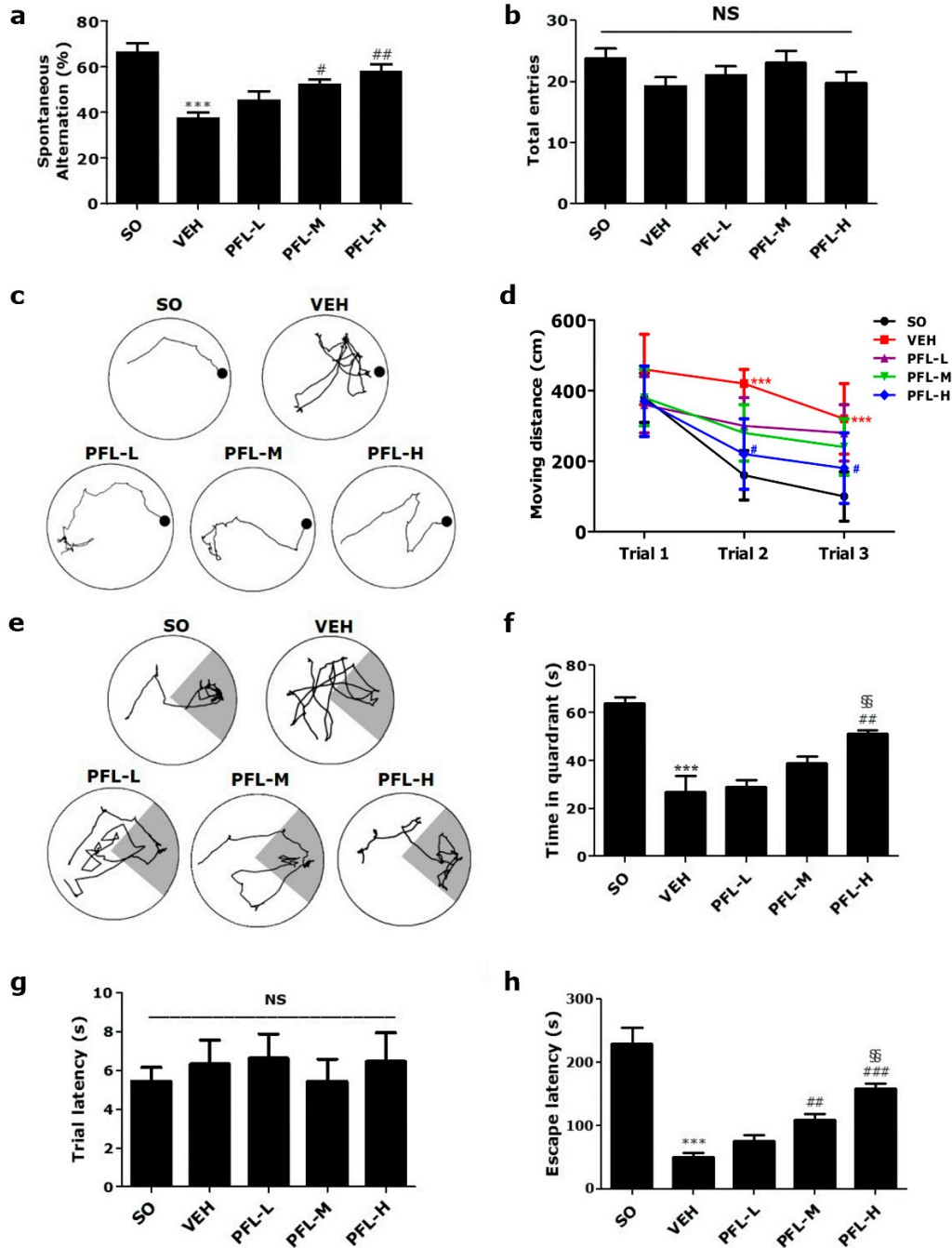

**Figure 3.** Effects of PFL on memory deficits in VaD rats. (**a**) Spontaneous alternation and (**b**) total arm entries of different groups were measured using the Y-maze test. Using the Barnes maze trial test session, (**c**) tracking plots made at the 3rd trial day and (**d**) time-dependent changes in the moved distance to the platform of different groups were obtained. Using the Barnes maze probe test session, (**e**) tracking plots and (**f**) time spent in the target quadrant of different groups were obtained. In the tracking plots, the locations of the platform and target quadrant are indicated as a black circle and dark area, respectively. (**g**) Trial latency and (**h**) escape latency in different groups were measured using the passive avoidance test. In all graphs, values are presented as the mean $\pm$ SEM (*** $p < 0.001$ vs. SO; # $p < 0.05$, ## $p < 0.01$, and ### $p < 0.001$ vs. VEH; §§ $p < 0.01$ vs. PFL-L; and NS, statistically not significant).

### 3.2. PFL Attenuates Deteriorations in Hippocampal Structures in the VaD Rats

Next, we investigated whether PFL intake could attenuate neuronal damage in the hippocampal structure, which is an essential feature in VaD patients [27]. For this, we used C-V staining to assess the effects of perioperative PFL treatment on the 2VO/H-induced loss of pyramidal neurons in the hippocampus, especially in the CA1 subregion. On POD 9, the VEH group showed a dramatic decrease in the number of viable hippocampal CA1 neurons compared to that of the SO group (*** $p < 0.001$; Figure 4a,b). The VEH group also showed the presence of apoptotic neurons, which were characterized by pyknotic nuclei and shrunken cytoplasm, in all three layers, i.e., the stratum oriens (S.O), pyramidale (S.P), and radiatum (S.R), of the CA1 region. However, the number of surviving neurons was significantly higher in both the PFL-M and PFL-H groups (## $p < 0.01$ and ### $p < 0.001$ vs. VEH), and this change was dose dependent (§§ $p < 0.01$ vs. PFL-L). Since neuroinflammation is essentially combined with VaD, we next quantified the extent of microglial and astrocytic activation by immunohistochemical detection of Iba1 and GFAP, which are used as markers for microglia and astrocytes, respectively. The number of Iba1$^+$ microglia in all three layers was approximately 30 times higher in the VEH group ($30.0 \pm 0.8$ vs. $1.1 \pm 0.1$; *** $p < 0.001$; Figure 4a,c) than the SO group. However, the number of activated microglia was significantly lower in the PFL-M group (## $p < 0.01$ vs. VEH), and this reduction was more exaggerated in the PFL-H group (### $p < 0.001$ vs. VEH). Contrary to expectations, there was no difference between the optical densities of GFAP$^+$ astrocytes in the PFL-treated groups and VEH group (Figure 4a,d; NS, statistically not significant). Collectively, these results suggest that PFL uptake could attenuate both VaD-associated neuronal loss and the associated microglial activation.

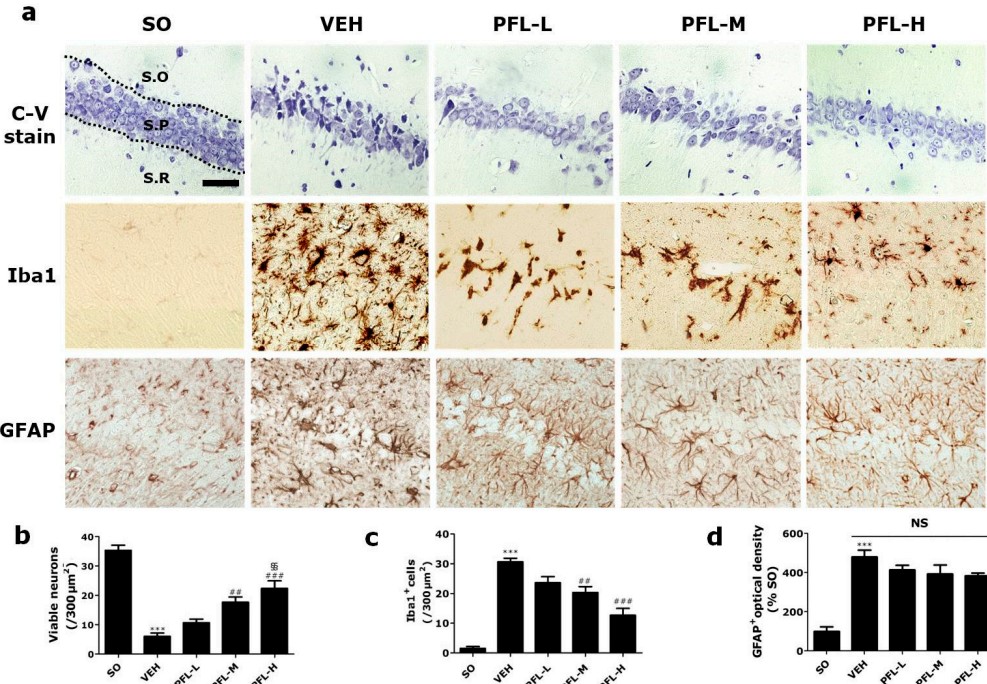

**Figure 4.** Effects of PFL on neuronal viability and neuroinflammation in the VaD rat hippocampal CA1. After cresyl violet staining (**a**, top row) and immunohistochemistry against Iba1 (**a**, middle row) and GFAP (a, bottom row), representative photographs of hippocampal CA1 were obtained, and the number of (**b**) viable neurons, (**c**) Iba1$^+$ microglia, and (**d**) optical density of GFAP$^+$ astrocytes were quantified, respectively. The region of interest was within a 300 μm width in the hippocampal CA1 region. In (**a**), the three layers of the hippocampus, i.e., stratum oriens, stratum pyramidale, and stratum radiatum, are indicated as "S.O", "S.P", and "S.R", respectively. Scale bar = 50 μm. In (**b**) to (**d**), values are presented as the mean ± SEM (*** $p < 0.001$ vs. SO; ## $p < 0.01$ and ### $p < 0.001$ vs. VEH; §§ $p < 0.01$ vs. PFL-L; NS, statistically not significant).

### 3.3. PFL Inhibits the Production of NO, TNF-α, and IL-6 in LPS-stimulated BV-2 Cells

To provide insight into the possible mechanisms underlying the PFL-mediated attenuation of microglial activation, the cytotoxic effect of PFL on BV-2 microglial cells was first evaluated in the presence or absence of LPS (1 μg/mL) using MTT assays. As shown in Figure 5a, PFL showed no cytotoxicity at all concentrations (50, 100, and 200 μg/mL) tested in this study, regardless of the presence of LPS. Accordingly, we used PFL at the indicated concentrations to further evaluate the inhibitory effects on the production of inflammatory mediators, such as NO, TNF-α, and IL-6, as shown in Figure 5b–d, respectively. When BV-2 cells were pretreated with PFL for 12 h, LPS-mediated production of NO, TNF-α, and IL-6 was significantly decreased. Among the dynamics, pretreating BV-2 cells with PFL showed a dose-dependent attenuation in LPS-induced IL-6 production (vs. $111.2 \pm 4.7$ in 50 μg/mL PFL-treated $5.8 \pm 3.2$ in 200 μg/mL PFL-treated; [#] $p < 0.05$; Figure 5d). Treatment with PFL did not affect the parameters mentioned above under the LPS-free conditions. Together, these results indicate that PFL suppressed NO, TNF-α, and IL-6 production in activated microglia.

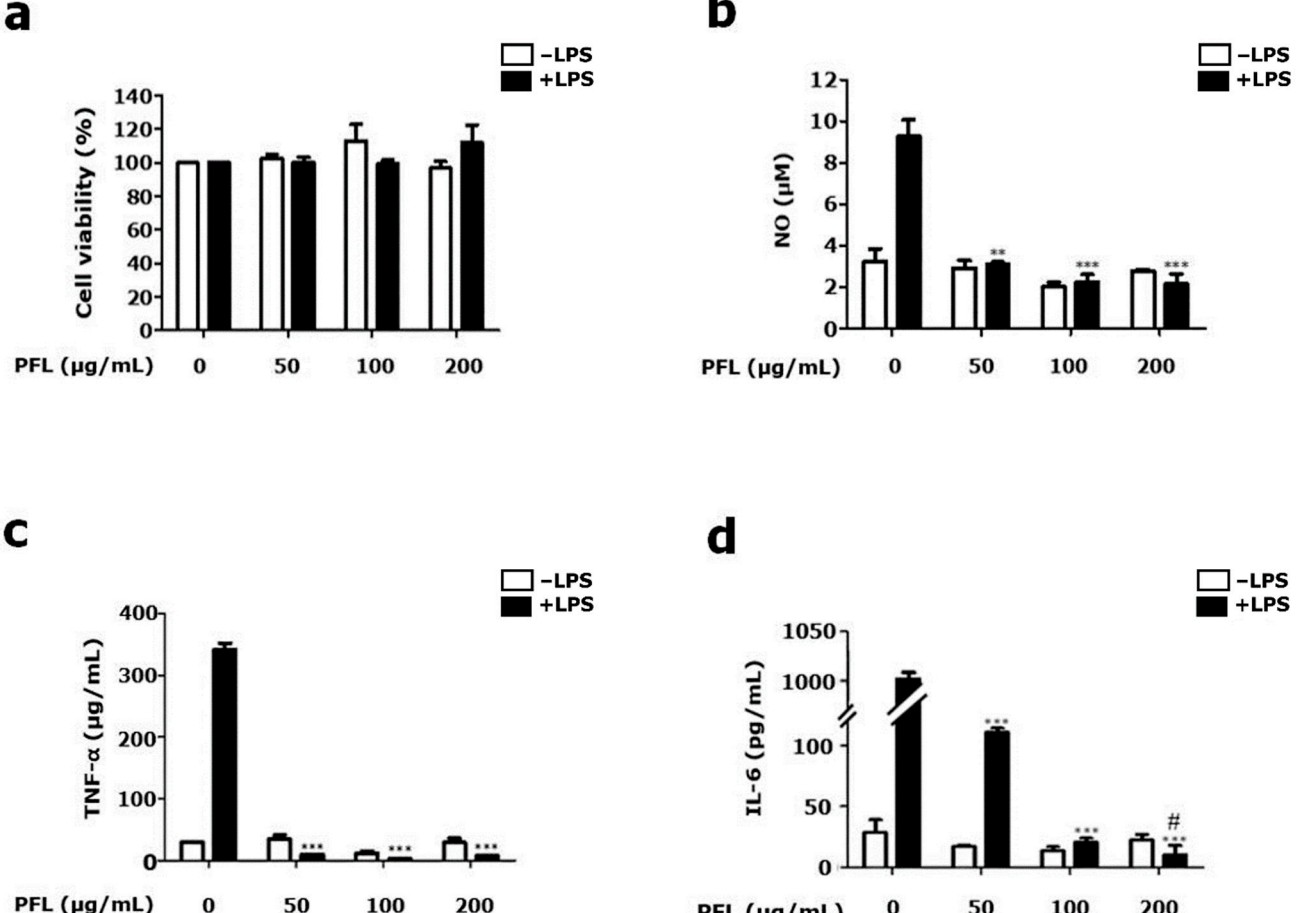

**Figure 5.** Effect of PFL on BV-2 microglial cell viability and NO, TNF-α, and IL-6 production. BV-2 cells were pretreated with the indicated doses of PFL for 12 h and further treated with 1 μg/mL lipopolysaccharide (LPS). After 24 h, (**a**) cell viability was determined by the methyl thiazolyl tetrazolium (MTT) assay. Using enzyme-linked immunosorbent assay (ELISA) kits, the contents of (**b**) NO, (**c**) TNF-α, and (**d**) IL-6 in culture media were determined. Values are presented as the mean $\pm$ SEM (** $p < 0.01$ and *** $p < 0.001$ vs. PFL-untreated; [#] $p < 0.05$ vs. 50 μg/mL PFL-treated).

### 3.4. PFL Inhibits MAPKs/NFκB/iNOS Signaling in LPS-Stimulated BV-2 Cells

Given that MAPKs activation and iNOS expression are involved in the production of diverse inflammatory mediators in microglia as a key upstream regulator and vice versa [28], we analyzed the phosphorylation of three kinds of MAPKs and the expression level of iNOS in LPS-stimulated BV-2 cells pretreated with the indicated concentrations of PFL. As shown in representative blot images (Figure 6a) and their quantification graphs (Figure 6b–d for p-p38, p-ERK1/2, and p-JNK, respectively), PFL pretreatment significantly suppressed the LPS-induced phosphorylation of all three MAPKs ([#] $p < 0.05$, [##] $p < 0.01$, and [###] $p < 0.001$ vs. LPS-only). In particular, in the JNK phosphorylation dynamics, there was an apparent dose-dependent attenuation ([§] $p < 0.05$ vs. 50 μg/mL PFL-treated; Figure 6d). Furthermore, PFL pretreatment significantly suppressed LPS-induced iNOS expression in a dose-dependent manner ([###] $p < 0.001$ vs. LPS-only; [§] $p < 0.05$ vs. 50 μg/mL PFL-treated; Figure 6a,e). As nuclear translocation-dependent activation of NFκB is reciprocally engaged in the activation of MAPKs and iNOS [9], we assessed the levels of nuclear NFκB (Figure 6a,f). The results indicated that the LPS-induced nuclear translocation of NFκB was significantly inhibited by pretreatment with 200 μg/mL PFL ([###] $p < 0.001$ vs. LPS-only). Collectively, these data suggest that inhibition of MAPKs/NFκB/iNOS signaling is responsible for the PFL-mediated attenuation of NO, TNF-α, and IL-6 production in activated microglia.

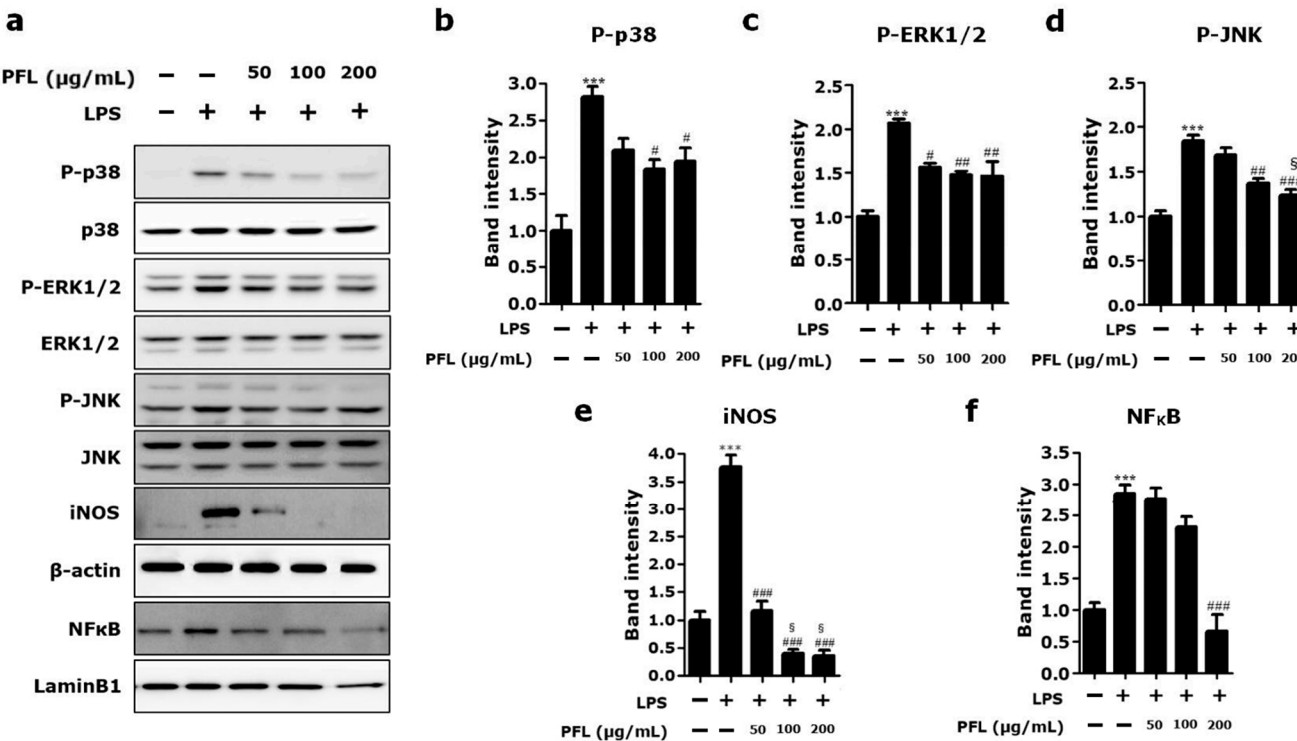

**Figure 6.** Effect of PFL on the expression of selected proteins in LPS-stimulated BV-2 cells. BV-2 cells were pretreated with the indicated doses of PFL for 12 h and further treated with 1 μg/mL lipopolysaccharide (LPS) for 24 h. By Western blot, (**a**) representative protein band images were obtained, and the amounts of phosphorylated MAPKs, including (**b**) p-p38, (**c**) p-ERK1/2, (**d**) p-JNK, (**e**) iNOS, and (**f**) nuclear NFκB, were semi-quantitatively assessed. The band intensities of p-p38, p-ERK1/2, p-JNK, and iNOS were normalized by β-actin, whereas nuclear NFκB intensity was normalized by LaminB1, each used as internal controls. Data were expressed as the fold of the controls and collected from at least three independent experiments. Values are presented as the mean ± SEM (*** $p < 0.001$ vs. LPS-untreated; [#] $p < 0.05$, [##] $p < 0.01$, and [###] $p < 0.001$ vs. LPS only-treated; [§] $p < 0.05$ vs. 50 μg/mL PFL-treated).

## 4. Discussion

In the present study, we investigated whether PFL intake might have therapeutic value against VaD. Using an in vivo model of VaD, we showed that PFL could ameliorate VaD-associated memory deficits, hippocampal neuronal death, and microglial activation. Furthermore, using an in vitro model of microglial activation, we showed that PFL could inhibit the production of proinflammatory mediators and the key upstream signaling pathways, that is, MAPK/NFκB/iNOS.

VaD is characterized by a progressive worsening of memory and other cognitive functions as a consequence of CVA [1]. The mechanisms that induce cognitive impairment in VaD are multifaceted, such as neuronal apoptosis, oxidative stress, and neurotransmitter abnormities [29]. Remarkably, many studies have mentioned that neuroinflammation is important in the development of cognitive impairment with VaD [30,31]. Upon the onset of neuroinflammation, microglia are activated and release proinflammatory mediators that favor the permeabilization of the blood–brain barrier [32,33]. The subsequent infiltration of peripheral leukocytes occurs inside the central nervous system. The inflammatory and neurotoxic mediators produced by infiltrated leukocytes exaggerate neuroinflammation, which culminates in a functional modification of hippocampal excitatory synapses and leads to cognitive impairment [34,35]. Therefore, there is a need for an unveiling of new therapeutic strategies that can counteract microglial activation.

In recent years, owing to the desire for safe, non-toxic, and brain-deliverable agents, there has been a growing interest, supported by a large number of experimental and epidemiological studies, in the protective effects of some herbal products with a natural origin in neurodegenerative diseases including VaD [36,37]. Such natural products often contain active phenolic substances endowed with potent antioxidative and anti-inflammatory properties. Recently, a series of studies focused on specific neuroprotective effects of some of those polyphenols derived from nutritional sources [38]. In this study, we revealed that PFL contains three major polyphenolic compounds, that is, scutellarin, apigenin-7-O-glucuronide, and rosmarinic acid, by HPLC. Among these, scutellarin was previously reported to rescue neurological deficits and restrict infarction volume in a focal stroke model, which was established by middle cerebral artery occlusion in rats [39]. Another study revealed that scutellarin could reduce neuronal damage in a global stroke model, which was established by bilateral CCA occlusion in rats, by modulating neurotransmitter levels and membrane channel activities [40]. Apigenin-7-O-glucuronide was reported to improve memory and learning deficits as well as reduce fibrillar amyloid deposits with lowered insoluble concentrations of $\beta$-amyloid peptide in transgenic AD mice [41]. Additionally, it was shown that apigenin-7-O-glucuronide caused the restoration of the cAMP-response element-binding protein (CREB)/brain-derived neurotrophic factor (BDNF) pathway, which is commonly deteriorated in AD [42]. Nneuroprotective actions of rosmarinic acid in rodent models of Parkinson's disease [43], AD [44], and amyotrophic lateral sclerosis [45] have been reported. One previous report revealed that rosmarinic acid-mediated neuroprotection involves stimulation of the antiapoptotic and antioxidant enzymatic defense system [46,47].

As shown above, when considering the accumulating evidence regarding the neuroprotective efficacies of the three polyphenols, it is reasonable that PFL, as a composite of these compounds, might be neuroprotective. However, to the best of our knowledge, there is a dearth of scientific information regarding the anti-VaD potential of PFL. This is due to, at least in part, the lack of unified and reproducible methods for the establishment of rodent VaD models [48]. At present, bilateral CCA (2-vessel) occlusion (2VO) in rats has been most commonly utilized as a rodent VaD model; incomplete interruption of CBF via the patent vertebral arteries (VAs) limits widespread usage as an experimental platform for drug screening [49]. In contrast to the issue surrounding 2VO, transient occlusion of bilateral CCA as well as bilateral VA (4-vessel occlusion; 4VO), another commonly utilized technique for modeling VaD, is highly invasive and is consequently associated with a high mortality rate, further increasing the number of animals necessary [50]. To

overcome the major drawbacks of 2VO and 4VO, we utilized 2VO/H, a relatively new technique, to generate the VaD model. We confirmed, by several preliminary trials, that this technique produced a reproducible injury to selectively vulnerable brain regions, such as the hippocampus, without a high mortality rate. The reliability of this model is attributed to the real-time and direct visualization of CBF changes during the ischemic period. An 8-min ischemic insult produced reproducible loss of hippocampal CA1 neurons, while less vulnerable brain regions were spared, thus yielding selective deterioration of hippocampus-dependent memory function without prefrontal cortex-dependent affective or motor deficits, as evidenced by this study (Figure 3b,g).

This study, for the first time, provides evidence of the anti-VaD efficacy of PFL, showing the attenuation of memory deficit and hippocampal neuronal death, the two essential features of VaD. By performing an in vitro experiment, we also report the PFL-mediated attenuation of microglial activation. However, major limitations of this study remain. First, we could not identify whether PFL-mediated neuroprotection is therapeutic, mainly acting after VaD onset, or preventive, acting prior to the insult. To address this issue, further studies using separate treatment protocols, that is, pretreatment versus posttreatment, are needed. Second, this study lacked detailed explanations regarding the predominant protective mechanisms underlying PFL-mediated anti-VaD effects in vivo. Although the possible modulation of microglial activation by PFL was focused upon in this study, we cannot rule out the possibility that PFL directly protects neurons against hypoxic damage, which can otherwise cause neuroinflammation. Considering that VaD involves complex pathological cascades, including a series of molecular and cellular events other than neuroinflammation, such as apoptosis [51], compromised neurogenesis [52], mitochondrial dysfunction [53], and ROS-associated oxidative damage [54], more advanced and well-designed studies should be performed in the future to explain the specific mechanisms underlying the PFL-mediated anti-VaD effect in detail.

## 5. Conclusions

PFL uptake diminished the structural and functional deteriorations exhibited in the VaD rat model established by 2VO/H. PFL pretreatment attenuated proinflammatory mediators including NO, TNF-$\alpha$, and IL-6, and downregulated the upstream MAPKs/NF$\kappa$B/iNOS pathway in LPS-stimulated BV-2 microglial cells. These results show that PFL has the potential to be a useful therapeutic strategy for VaD.

**Author Contributions:** Conceptualization, S.-Y.H.; validation, H.-B.K.; investigation, S.-H.U.; methodology, D.-K.K., N.-S.L. and Y.-G.J.; resources, D.-S.K., N.-Y.S. and I.-J.H.; data curation, Y.-C.Y.; writing—original draft preparation, S.-H.K.; writing—review and editing, S.-Y.H.; supervision and funding acquisition, S.-Y.H. All authors have read and agreed to the published version of the manuscript.

**Funding:** This work was partly supported by the Industry-Academy Collabo Program (S3101908) funded by the Ministry of SMEs and Startups (MSS, Korea) and the Basic Science Research Program (NRF-2019R1C1C1002294) funded by the Ministry of Education, Science, and Technology (MEST, Korea).

**Institutional Review Board Statement:** Animal experiments in this study were approved by the Institutional Animal Care and Use Committee of Konyang University (Daejeon, Korea; approval code, P-20–31-A-01; approval date, 1 March 2021).

**Informed Consent Statement:** Not applicable.

**Data Availability Statement:** Not applicable.

**Conflicts of Interest:** The authors declare no conflict of interest.

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
