# Peer review of "Perilla frutescens Leaf Extract Attenuates Vascular Dementia-Associated Memory Deficits, Neuronal Damages, and Microglial Activation"

_cimb, doi:10.3390/cimb44010019_

Round 1

Reviewer 1 Report

The manuscript "Perilla frutescens Leaf Extract Attenuates Vascular Dementia-associated Memory Deficits, Neuronal Damages, and Microglial Activation" contains specific information on a possible therapeutic treatment in vivo in rats. The authors demonstrate specific results concerning the potential to use as a specific therapeutic treatment. There are some specific annotations regarding figures. There is one thing for specific understanding in Fig. 2A, B, and C, please present the most amplitude figure, because is important to understand the in vivo experiment setup. It needs an ample presentation figure or separated in different planes for example A and B and C and D. Change to capital letters in the legend of figure 3. As well, change to capital letters but in figure 4, or unified all for capital letters or lower-case letters. 

Reviewer 2 Report

The manuscript entitled “Perilla frutescens Leaf Extract Attenuates Vascular Dementia-2 associated Memory Deficits, Neuronal Damages, and Micro-3 glial Activation” explains the impact of supplementation PFLE using the rat model. We appreciate the study, and please revise the manuscript as per the comments below.

Comments

Line 72. Please change as “P. frutescens” not “the P. frutescens

Line 83 and 98. Scientific should be in italicize, e.g., Escherichia coli

Line 115. The gradient was 0-3 min, 10-10% B; is this 1-10%. Please clarify.

Line 161. Please italicize “in Vivo”

Line 328. If the data has not been used for other papers, please show your data in this paper.

Figures: Please create color figures for better clarity, especially Figure 3D.
